# In Vitro α-Amylase and α-Glucosidase Inhibitory Effects, Antioxidant Activities, and Lutein Content of Nine Different Cultivars of Marigold Flowers (*Tagetes* spp.)

**DOI:** 10.3390/molecules28083314

**Published:** 2023-04-08

**Authors:** Wason Parklak, Sakaewan Ounjaijean, Kanokwan Kulprachakarn, Kongsak Boonyapranai

**Affiliations:** 1Research Center for Non-Infectious Diseases and Environmental Health, Research Institute for Health Sciences, Chiang Mai University, Chiang Mai 50200, Thailand; toon.wason@gmail.com (W.P.); sakaewan.o@cmu.ac.th (S.O.); k_kulprachakarn@hotmail.com (K.K.); 2School of Health Sciences Research, Research Institute for Health Sciences, Chiang Mai University, Chiang Mai 50200, Thailand

**Keywords:** marigolds, bioactive compounds, lutein, antioxidant activity, α-amylase, α-glucosidase

## Abstract

Marigolds (*Tagetes* spp.) are major sources of bioactive compounds. The flowers are used to treat a variety of illnesses and have both antioxidant and antidiabetic effects. However, marigolds exhibit a wide range of genetic variations. Because of this, both the bioactive compounds and biological activities of the plants differ between cultivars. In the present study, nine marigold cultivars grown in Thailand were evaluated for their bioactive compound content, as well as for their antioxidant and antidiabetic activities, using spectrophotometric methods. The results showed that the Sara Orange cultivar possessed the highest total carotenoid content (431.63 mg/100 g). However, Nata 001 (NT1) had the highest amount of total phenolic compounds (161.17 mg GAE/g), flavonoids (20.05 mg QE/g), and lutein (7.83 mg/g), respectively. NT1 exhibited strong activities against the DPPH radical and ABTS radical cation, and had the highest FRAP value as well. Moreover, NT1 demonstrated the most significant (*p* < 0.05) α-amylase and α-glucosidase inhibitory effects (IC_50_ values of 2.57 and 3.12 mg/mL, respectively). The nine marigold cultivars had reasonable correlations between lutein content and the capacity to inhibit α-amylase and α-glucosidase activities. Hence, NT1 may be a good source of lutein; it may also be beneficial in both functional food production and medical applications.

## 1. Introduction

According to the International Diabetes Federation, the number of individuals diagnosed with diabetes globally in 2017 was 425 million; this figure was projected to increase to 642 million by 2050, with more than 60 percent of those affected residing in Asia [1]. Diabetes is a set of metabolic disorders defined by persistent hyperglycemia caused by insufficient insulin activity or production [2]. All varieties of diabetes are characterized by elevated blood sugar levels before and after eating, as well as a relative insulin deficiency. Individuals with diabetes are at significant risk for a variety of life-threatening health conditions. Hyperglycemia in uncontrolled diabetes mellitus causes chronic microvascular and macrovascular complications, such as retinopathy, neuropathy, nephropathy, weight gain, and atherosclerosis [1,2]. It is also linked to the accelerated development of atherosclerotic macrovascular disease affecting the arteries of the heart, brain, and extremities [3]. Recent studies have demonstrated that a high postprandial plasma glucose level is more dangerous than a high fasting blood glucose level. To decrease complications and death from diabetes, it is essential to regulate post-meal blood glucose levels [3]. One method for treating diabetes is to reduce postprandial glycemia by blocking the enzymes responsible for carbohydrate hydrolysis, such as α-amylase and α-glucosidase [4]. Inhibitors that block carbohydrate-degrading enzymes like α-amylase and α-glucosidase have been shown to be effective in lowering postprandial blood sugar levels in people with diabetes [5]. Nevertheless, these medications may induce hypoglycemia, liver problems, lactic acidosis, and diarrhea when used in larger quantities [4]. In addition to existing therapy alternatives, herbal medications containing bioactive compounds are a safe and effective alternative to conventional diabetes therapies [6,7].

Phytochemicals, the natural bioactive compounds of plants, appear to play a major role in both plant and human health [8]. Many bioactive compounds, including phenolic compounds (flavonoids, carotenoids, etc.), are present in the majority of plant species and perform numerous crucial ecological activities [9]. Phenolic compounds form a wide collection of naturally occurring organic molecules found in many plant morphological structures. They possess potent antioxidative characteristics that protect the body’s defensive mechanisms from the harmful impacts of free radicals [9]. Isoprenoids, which comprise flavonoids and carotenoids, are another category of secondary metabolites of plants that have health-promoting effects [8]. Flavonoids and carotenoids are the predominant flower pigments in most plants. They possess antioxidative, anti-inflammatory, antidiabetic, and anti-hyperglycemic properties [8,9,10,11,12].

Marigolds (*Tagetes* spp.) are regarded as the richest source of natural pigments among flowers; pigments are derived from flower petals and traded worldwide [13]. The two key pigment groups found in marigolds are flavonoids and carotenoids [14,15,16]. Specifically, lutein ester carotenoids have been identified as the predominant pigments in marigold petals [17,18]. These pigments are regularly found in both edible and inedible plants, and they have been credited with many biological activities, including antioxidant activity [8,9,17,18]. In addition, previous research has demonstrated the preventive effect of both crude and lutein marigold extract on hyperglycemia in diabetic rodent models [19,20]. Marigolds have a wide genetic diversity and are easily crossbred; there are at least one hundred distinct varieties of marigolds in the world. As a result, the quantity and composition of bioactive components vary between cultivars [21,22]. Therefore, the objective of the current study was to assess the bioactive compounds, antioxidant activities, and in vitro α-amylase and α-glucosidase inhibitory effects of nine different marigold cultivars, including Rocco Deep Gold (RDG), Golden King 001 (GK1), Yellow Queen 002 (YQ2), Nata 001 (NT1), Amari Deep Gold (ADG), Sara Deep Gold (SDG), Sara Orange (SO), Amari (AM), and Angka (AK) cultivated at an organic farm in the Mae Rim district, Chiang Mai province, Thailand.

## 2. Results

### 2.1. Bioactive Compound Contents in Nine Marigold Cultivars

The bioactive compound profiles (total phenolic, flavonoid, and carotenoid contents, respectively) for nine marigold cultivars are listed in Table 1. The values of phenolic content were found to range from 94.58 to 161.17 mg GAE/g dry weight. The highest phenolic compound content among the tested samples was found in NT1 (161.17 mg GAE/g dry weight), followed by GK1 (134.59 mg GAE/g dry weight) and YQ2 (131.14 mg GAE/g dry weight). The lowest total phenolic content was detected in SO and SDG (103.73 and 94.58 mg GAE/g dry weight, respectively).

The total flavonoid content ranged from 12.67 to 20.05 mg QE/g petal dry weight among the various marigold cultivars. NT1 had the highest total flavonoid content (20.05 mg QE/g dry weight), followed by YQ2, GK1, and ADG (17.92, 16.49, and 16.32 mg QE/g dry weight, respectively). However, AM, AK, RDG, SDG, and SO exhibited the lowest total flavonoid content (12.67, 13.59, 13.72, 13.73, and 14.36 mg QE/g dry weight, respectively).

Among the nine marigold cultivars studied, total carotenoid content showed high variation, ranging from 53.80 to 431.63 mg/100 g dry weight. SO had the highest total carotenoid content (431.63 mg/100 g dry weight), while the lowest content was recorded in YQ2 (53.80 mg/100 g dry weight).

The HPLC chromatogram of the lutein standard and marigold petal sample is shown in Figure 1; the lutein content for each of the nine marigold cultivars is shown in Figure 2. The lutein content ranged between 1.82 and 7.82 mg/g dry weight. NT1 possessed the highest lutein content (7.82 mg/g dry weight), followed by YQ2 (4.29 mg/g dry weight). In contrast, ADG, SO, SDG, and AM had the lowest lutein content (1.56, 1.82, 1.89, and 1.95 mg/g dry weight, respectively).

### 2.2. Antioxidant Activities of Nine Marigold Cultivars

The antioxidant activity results (obtained using DPPH, ABTS, and FRAP assay methods) of nine marigold cultivars are presented in Table 2. In the DPPH assays, the percentage of inhibition in DPPH free radical ranged from 53.14% to 67.23% at a 4 mg/mL concentration. YQ2 and NT1 demonstrated the highest percentages of inhibition (67.23% and 67.01%, respectively), followed by AK, ADG, and GK1 (61.05%, 59.53%, and 59.11%, respectively), whereas SDG and AM had the lowest percentages of antioxidant activity (53.14% and 53.30%, respectively). The percentage of inhibition in ABTS radical cation ranged from 67.01% to 90.41%. At a 4 mg/mL concentration, NT1 and YQ2 displayed the highest percentages of inhibition when compared with each other cultivars (90.41% and 89.57%, respectively), while the lowest percentages were found in GK1, AK, and RDG at the same concentration (67.01%, 68.69%, and 69.48%, respectively). In the FRAP assays, the antioxidant capacity level of the samples ranged from 330.31 to 392.30 µmol Trolox per gram of dry marigold petal. The highest FRAP value was found in the NT1 cultivar (392.30 µmol Trolox/g dry marigold petal). However, SDG, RDG, AK, and AM possessed the lowest FRAP values of 330.31, 330.75, 333.69, and 335.03 µmol Trolox/g dry marigold petal, respectively.

### 2.3. In Vitro α-Amylase and α-Glucosidase Inhibitory Effects of Nine Marigold Cultivars

The effect of the nine marigold cultivars on the inhibitory activity of α-amylase and α-glucosidase is shown in Figure 3. The standard acarbose was used as an antihyperglycemic agent to compare the inhibitory effects of various samples. The inhibitory effect of acarbose (0.04–5 mg/mL) against α-amylase and α-glucosidase exhibited dose-dependent activity (r^2^ = 0.9552 and 0.9916, respectively). At a concentration of 5 mg/mL, the inhibition of α-amylase (presented as IC_50_ values) among the nine marigold cultivars ranged from 2.37 to 4.87 mg/mL. YQ2 and NT1 demonstrated the most significant α-amylase inhibition (IC_50_ values of 2.37 and 2.57 mg/mL, respectively), while SO exhibited the least inhibitory action (IC_50_ value of 4.87 mg/mL). In the case of α-glucosidase inhibition, the IC_50_ values ranged between 3.12 and 7.40 mg/mL The inhibitory effect of NT1 (IC_50_ value of 3.12 mg/mL) on α-glucosidase was substantially (*p* < 0.05) higher than that of all other cultivars. In contrast, SDG had the lowest inhibitory activity (IC_50_ value of 7.40 mg/mL).

### 2.4. Correlation Analysis

Pearson’s correlation coefficients (*r*) were used to analyze the direction and magnitude of associations between bioactive compound contents, antioxidant activities, and the α-amylase and α-glucosidase inhibitory potential of nine marigold cultivars (as shown in Table 3). The analysis revealed highly positive correlations between total phenolic, total flavonoid, and lutein content with all antioxidant assays (DPPH, ABTS, and FRAP assays) and the α-amylase and α-glucosidase inhibitory potential of all marigold cultivars (*r* values ranged from 0.498 to 0.872). In contrast, significant negative correlations were observed between total carotenoid content and both the DPPH (*r* = −0.685) and FRAP (*r* = −0.500) assays, as well as between total carotenoid content and the inhibition of α-amylase and α-glucosidase by all marigold cultivars (*r* = −0.866 and −0.554, respectively).

## 3. Discussion

As a key source of natural medicines, plants are known to have high levels of phenolic compounds and powerful antioxidant (or free radical scavenging) action [10]. Many epidemiological studies indicate that bioactive compounds play a preventive function in health and diseases [9,10]. Marigolds, which originated in Mexico, are biologically varied, flowering plants found in temperate regions. Given the marigold’s ability to cross widely, there are at least one hundred types of marigolds in the world. Hence, the content of bioactive components differs between cultivars [21,22]. According to previous studies, the principal bioactive compounds in marigold petal extracts include phenolics and carotenoids. Gallic acid and quercetin were the predominant phenolics in these extracts, but lutein is the predominant carotenoid in marigold petals [23,24]. Of the marigold cultivars tested in this study, the total carotenoid concentration of the examined marigold cultivars ranged from 53.80 mg/100 g dry weight to 431.63 mg/100 g dry weight, exhibiting a wide variance. This is congruent with the findings of Akshaya et al. [24]: they determined that the total carotenoid concentration of Indian marigold petals varied between 19.61 mg/100 g and 525.68 mg/100 g. The present results demonstrated that SO produced the highest total carotenoid content. In contrast, YQ2 and NT1 produced the lowest levels. In addition, this study demonstrated that the total carotenoid content changed with the intensity of the plant pigment. Carotenoids were abundant in orange cultivars such as SO or AM, while yellow cultivars like YQ2 and NT1 had a low carotenoid content. These total carotenoid content findings are consistent with those of Gregory et al. [25] and Kasemsap et al. [26]. Marigolds with a deep orange color, as opposed to a lighter orange or yellow, contained a greater concentration of carotenoids, and they were also particularly well-suited for carotenoid pigment extraction for commercial application. According to other research, the major carotenoid in edible flowers is lutein, which is responsible for their yellow color [27,28,29]. This may explain the extremely high concentration of orange-colored carotenoids in marigold petals; several researchers have found zeaxanthin and β-carotene to be the most abundant components [30]. Furthermore, research indicates that carotenoid variability depends on botanical origin, environmental variables within species, the choice of plant sections studied, and growing environment [27,28,29]. Consistent with previous reports, this study found that yellow cultivars, including NT1, had the highest levels of total phenolic compounds, flavonoids, and lutein. In contrast, orange-colored marigold cultivars were found to contain low levels of total phenolic compounds, flavonoids, and lutein; this finding contradicted those of several other studies [27,28,29]. This may have resulted from choosing plants with varying origins. However, positive associations between lutein and antioxidant activities were found in all nine marigold cultivars studied.

The antioxidant activity of phytochemical compounds is essential for both regulating the redox state in the body and decreasing disease-related damage. Many studies have established that bioactive chemicals derived from natural sources possess antioxidative, anti-inflammatory, antidiabetic, and anti-hyperglycemic properties [8,9,10,11,12]. In this study, the antioxidant activity of nine different marigold cultivars was evaluated using three different assays (DPPH, FRAP, and ABTS); the NT1 cultivar appeared to act as a high reducing agent (FRAP assay) and displayed stronger activity against DPPH radical and ABTS radical cation than most of the other cultivars. Kaisoon et al. [31] reported similar findings: They examined the antioxidant properties of 12 edible flowers and found that *Tagetes erecta* inhibited DPPH by the highest percentage, 85.70%. Similar to the current findings, they observed *Tagetes erecta* FRAP values between 329.4 and 609.2 µmol Trolox/g and 94.3% DPPH radical scavenging activity [32]. Moreover, antioxidant activity in *Tagetes erecta* was also found by Munira [33] and Pratheesh et al. [34].

Inhibitors of amylase and glucosidase are sometimes referred to as starch blockers because they include chemicals that inhibit the absorption of dietary starch by the body. Starch is a complex carbohydrate that cannot be absorbed without first being degraded by amylase and other secondary enzymes [4,5]. Amylase inhibitors with a high concentration have demonstrated the ability to reduce glucose absorption in humans [3]. Current research indicates that bioactive compounds, including phenolics, play a function in amylase and glucosidase suppression and hence have the potential to aid in the treatment of type 2 diabetes [6,7]. In the present study, the inhibitory effects of diabetes mellitus (DM)-related enzymes α-amylase and α-glucosidase were evaluated in the nine marigold cultivars using acarbose, a typical antihyperglycemic drug. NT1 inhibited both α-amylase and α-glucosidase to a greater extent than all other cultivars. Values of IC_50_ against α-amylase and α-glucosidase were found to be in the range of 2.37–4.87 mg/mL and 3.12–7.40 mg/mL, respectively. Nowicka and Wojdyło [35] studied the α-amylase and α-glucosidase inhibitory effects of 16 edible flowers and discovered that yellow-petaled flowers had the greatest anti-hyperglycemic potential. Consistent with that study, the current research found that yellow-colored NT1 cultivars demonstrated the most significant inhibition of α-amylase and α-glucosidase activity. Moreover, highly positive correlations were observed between the inhibitory potential of both α-amylase and α-glucosidase and lutein content. The association between lutein content and inhibitory potential is consistent with the results of both Kusmiati et al. [19] and Rodda et al. [20]. Both crude and lutein extracts from *Tagetes erecta* have demonstrated a significant effect on blood glucose levels in diabetic rodent models. This implies that the lutein extract from marigold petals may block carbohydrate-degrading enzymes like α-amylase and α-glucosidase, which may in turn result in lower postprandial blood sugar levels in rodents with diabetes.

## 4. Materials and Methods

### 4.1. Preparation of Marigold Petals

Win All Chocolate Co., LTD. provided nine varieties of marigold flower petals from their organic farm (Chiang Mai Province, Thailand). Descriptions of the nine varieties of marigold flowers are given in Table 4. Nine cultivars of dried marigold petals were extracted with 95% ethanol by continuously shaking at 120 rpm and 25 °C for 24 h. The sample was subsequently filtered using a 0.45 µm membrane, and the filtrate was kept in a refrigerator at 4 °C in the absence of light.

### 4.2. Determination of Phytochemical Contents

Determination of total phenolic content: A Folin–Ciocalteu assay was used to determine the total phenolic content of each extract [36]. Briefly, 0.5 mL of crude extract was combined for 4 min with 2.5 mL of a 10% Folin–Ciocalteu reagent, followed by the addition of 2 mL of 7.5% (w/v) sodium carbonate. After keeping the mixture in the dark for 30 min, its absorbance at 765 nm (Shimadzu, Kyoto, Japan) was measured. The total phenolic content was determined using a calibration curve, and the findings were represented in milligrams of gallic acid equivalent (GAE) per gram of dry weight.

Determination of total flavonoid content: The total flavonoid concentration of each extract was measured using the aluminum chloride colorimetric technique [37]. Briefly, 0.5 mL of extract was combined with 0.1 mL of 10% (w/v) aluminum chloride and 0.1 mL of 1-M potassium acetate. Thereafter, 4.3 mL of distilled water was added to the mixture. After allowing the mixture to stand for 30 min, its absorbance was measured at 415 nm (Shimadzu, Kyoto, Japan). The total flavonoid concentration was determined using a calibration curve, and the result was represented in milligrams of QE per gram of dry weight.

Determination of total carotenoid content: Total carotenoids were extracted following Akshaya’s specified technique [24]. Briefly, 0.1 g extract was weighed and coarsely pulverized in acetone using a mortar and pestle until the residue became colorless. A separating funnel was used to separate carotenoid pigments. Then, the carotenoid extract was transferred to a separating funnel, followed by the addition of petroleum ether and 10% Na_2_SO_4_. Next, the funnel was spun to separate the carotenoid layer, and the carotenoids were gathered in a volumetric flask. The procedure was repeated until the remaining extract was colorless. The absorbance was measured spectrophotometrically at 452 nm (Shimadzu, Kyoto, Japan). Total carotenes were computed using the following formula:


(1)
$$\mathrm{Total}\;\mathrm{carotenoids}\;(\mathrm{mg}/100\;g)=\frac{3.87\times \mathrm{abs}\times \mathrm{volume}\;\mathrm{makeup}\times \mathrm{dilution}\;\mathrm{factor}\times 100}{\mathrm{weight}\;\mathrm{of}\;\mathrm{sample}\;(g)\times 1000}$$


Determination of lutein content: Each marigold petal extract was combined with an extractant (hexane:acetone; 10:6). The technique described by Piccaglia et al. for quantifying lutein was utilized [13]. In brief, 20 µL of the diluted extract was injected into the HPLC column, which was a The Zorbax SB C18 (15 cm × 4.6 mm i.d., 5 µm) reversed-phase column (Agilent technologies, Santa Clara, CA, USA). The peaks were detected by comparing both the retention time and spectrum to the lutein standard. The amount of lutein was given as mg/g of dry weight.

### 4.3. Determination of Antioxidant Activities

DPPH radical scavenging assay: Antioxidant activity was measured using 2,2-diphenyl-1-picrylhydrazyl hydrate (DPPH) free radical scavenging activity; this method was adapted from Kulprachakarn et al. [38]. Briefly, 200 µL of each sample was combined with 200 µL of a 0.4-mM DPPH solution and then incubated in the dark at room temperature for 30 min. Trolox acted as a positive control. The absorbance was then measured using a microplate reader at a wavelength of 517 nm (Shimadzu, Kyoto, Japan). The capacity of DPPH radical scavenging was reported as a percentage of inhibition, which was computed using the equation below.


Percentage of inhibition (%) = (Abs_control_ − Abs_sample_)/Abs_control_ × 100(2)


ABTS radical cation scavenging assay: Potassium persulfate solution 2.45 mM and 7 mL of 2,2′-azino-bis(3-ethylbenzothiazoline-6-sulfonic acid) (ABTS) solution were utilized as stock solutions. The working solution was then created by combining the two stock solutions in equal amounts and letting them react in the dark for 12–16 h at room temperature. Using a spectrophotometer, the solution was diluted with distilled water to achieve an absorbance of 0.700 ± 0.02 units at 734 nm. Briefly, 10 µL of extract was combined with 1 mL of ABTS solution and then kept in the dark at room temperature for 6 min. At 734 nm, the absorbance of the combination was measured with a spectrophotometer (Shimadzu, Kyoto, Japan) [38]. The capacity of ABTS radical cation scavenging was reported as a percentage of inhibition and computed using Equation (2).

FRAP Assay: A ferric-reducing antioxidant power (FRAP) test, adapted from Kulprachakarn et al. [36], was conducted for each extract. The FRAP reagent was prepared by combining 300-mM acetate buffer and 10-mM 2,4,6 tripyridyl-s-triazine (TPTZ) in 40-mM HCl and 20-mM FeCl_3_ in a 10:1:1 ratio. Microplate readers measured reacted samples at 595 nm (BioTek Instruments Inc., Winooski, VT, USA), and ascorbic acid (1 mg/mL) was used as a positive control. A calibration curve was generated using Trolox, with results expressed as µmol Trolox per gram of dry weight.

### 4.4. In Vitro Antidiabetic Activity

Alpha—amylase inhibition activity: p-nitrophenyl-α-D-glucopyranoside was digested to conduct the yeast α-glucosidase (G0660, Sigma-Aldrich, Bangkok, Thailand) enzyme inhibition experiment described by Tadera et al. [12]. The sample solution (2 µL dissolved in DMSO) was combined with 0.5 U/mL α-glucosidase (40 µL) in 120 µL of 0.1 M phosphate buffer (pH 7.0). Following a 5-min preincubation, 40 µL of a 5 mM p-nitrophenyl-α-D-glucopyranoside solution was added, followed by a 30-min incubation at 37 °C. Using a microplate reader, the absorbance of released 4-nitrophenol was determined at 405 nm (BioTek Instruments Inc., Winooski, VT, USA). The positive control was acarbose.

Alpha—glucosidase inhibitory activity: The enzyme inhibitory activity of porcine pancreatic α-amylase (A3176, Sigma-Aldrich) was determined using a technique described by Tadera et al. [12] with minor changes. 2-chloro-4-nitrophenyl-α-D-maltotrioside (93834, Sigma-Aldrich) was dissolved in phosphate buffer to create substrate (pH 7.0). Next, 2 µL of the DMSO-dissolved sample and 50 µL of 0.5 unit/mL α-amylase were combined in 100 µL of 0.1 M phosphate buffer (pH 7.0). Following a preincubation period of 5 min, 50 µL of substrate solution was added and the solution was incubated at 37 °C for 15 min. At 405 nm, the absorbances were determined (BioTek Instruments Inc., USA). Acarbose was employed as a positive control.

### 4.5. Statistical Analysis

Each cultivar’s marigold extracts were examined in triplicate, and the mean values were reported. Using SPSS (Version 16; IBM, Armonk, NY, USA), analysis of variance (ANOVA) and Duncan’s multiple-range tests were applied to compare the samples. Pearson’s *r* was used to examine linear correlations.

## 5. Conclusions

This study evaluated both the bioactive component profiles and antioxidant and antidiabetic efficacy of nine distinct marigold varieties. The NT1 cultivar was shown to possess the highest total phenolic compounds, flavonoid content, and lutein content, respectively. In addition, this variety excelled in free-radical scavenging and exhibited the highest α-amylase and α-glucosidase inhibitory activity. These results suggest that the NT1 cultivar may be useful in preventing and treating hyperglycemia and other complications associated with free radicals. Further in vivo studies using animal models are needed to confirm the in vitro findings reported in this study.

## Figures and Tables

**Figure 1 molecules-28-03314-f001:**
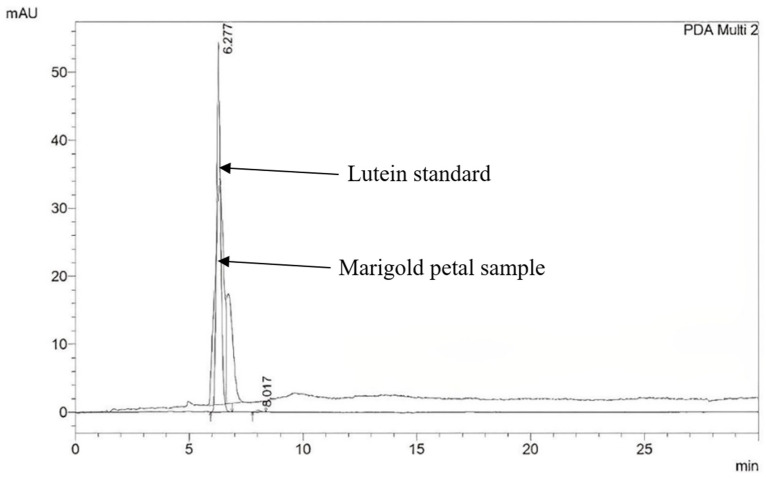
HPLC chromatogram of the lutein standard and marigold petal sample (NT1).

**Figure 2 molecules-28-03314-f002:**
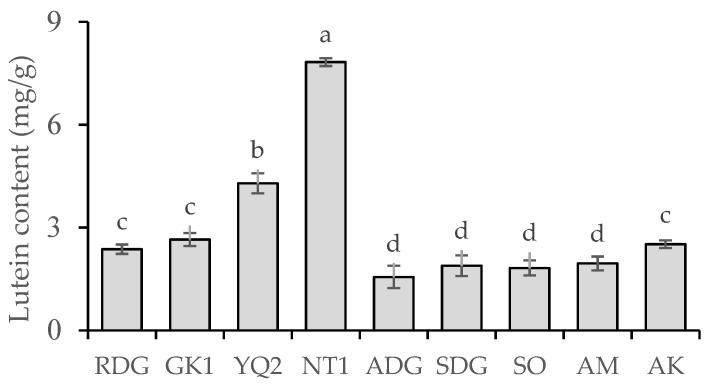
Lutein content of nine marigold cultivars. Mean values with varying letters are significantly different (*p*-value < 0.05). Rocco Deep Gold, RDG; Golden King 001, GK1; Yellow Queen 002, YQ2; Nata 001, NT1; Amari Deep Gold, ADG; Sara Deep Gold, SDG; Sara Orange, SO; Amari, AM; Angka, AK.

**Figure 3 molecules-28-03314-f003:**
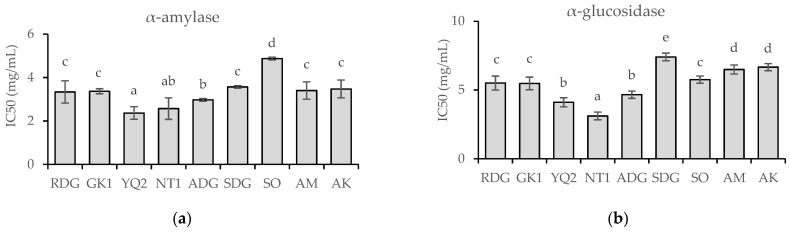
Half-maximal inhibitory concentration (IC_50_) values for α-amylase (**a**) and α-glucosidase (**b**) in nine marigold cultivars. Mean values with varying letters are significantly different (*p-*value < 0.05).

**Table 1 molecules-28-03314-t001:** Total phenolic, flavonoid, and carotenoid content of nine marigold cultivars.

Cultivar Name	Total Phenolic Content (mg GAE/g)	Total Flavonoid Content (mg QE/g)	Total Carotenoid Content (mg/100 g)
RDG	118.95 ± 7.24 ^cd^	13.72 ± 1.39 ^c^	212.21 ± 5.22 ^d^
GK1	134.59 ± 5.99 ^b^	16.49 ± 1.05 ^b^	133.26 ± 4.84 ^f^
YQ2	131.14 ± 2.07 ^bc^	17.92 ± 1.38 ^b^	53.80 ± 1.56 ^i^
NT1	161.17 ± 9.77 ^a^	20.05 ± 0.93 ^a^	58.57 ± 1.16 ^h^
ADG	116.14 ± 7.68 ^de^	16.32 ± 0.88 ^b^	91.46 ± 1.75 ^g^
SDG	94.58 ± 4.48 ^f^	13.73 ± 0.61 ^c^	225.49 ± 4.10 ^c^
SO	103.73 ± 10.62 ^ef^	14.36 ± 1.31 ^c^	431.63 ± 11.23 ^a^
AM	121.21 ± 3.91 ^cd^	12.67 ± 0.61 ^c^	267.80 ± 5.24 ^b^
AK	112.98 ± 7.06 ^de^	13.59 ± 0.93 ^c^	145.90 ± 2.32 ^e^

Values of mean ± SD (*n* = 3) followed by different lowercase superscript letters are significantly different at *p*-value < 0.05 according to Duncan’s multiple range test.

**Table 2 molecules-28-03314-t002:** The percent inhibition of DPPH radical and ABTS radical cation and ferric reducing antioxidant power (FRAP) value of nine marigold cultivars.

Cultivar Name	DPPH (% Inhibition)	ABTS (% Inhibition)	FRAP (µmol Trolox/g)
RDG	55.55 ± 0.96 ^c^	69.48 ± 2.66 ^de^	330.75 ± 0.65 ^d^
GK1	59.11 ± 1.22 ^b^	67.01 ± 1.73 ^e^	350.86 ± 0.86 ^c^
YQ2	67.23 ± 1.39 ^a^	89.57 ± 1.19 ^a^	361.91 ± 3.33 ^b^
NT1	67.01 ± 1.52 ^a^	90.41 ± 0.92 ^a^	392.30 ± 3.75 ^a^
ADG	59.53 ± 1.13 ^b^	75.26 ± 1.38 ^bc^	359.29 ± 3.83 ^b^
SDG	53.14 ± 1.81 ^d^	73.66 ± 2.61 ^bc^	330.31 ± 1.74 ^d^
SO	56.85 ± 1.37 ^c^	77.04 ± 1.33 ^b^	349.27 ± 4.98 ^c^
AM	53.30 ± 0.98 ^d^	72.07 ± 2.56 ^cd^	335.03 ± 4.25 ^d^
AK	61.05 ± 0.22 ^b^	68.69 ± 1.69 ^de^	333.69 ± 4.66 ^d^

Values of mean ± SD (*n* = 3) followed by different lowercase superscript letters are significantly different at *p*-value < 0.05 according to Duncan’s multiple range test.

**Table 3 molecules-28-03314-t003:** Linear correlation coefficients (*r*) between phytochemical content, antioxidant activity, and the percentage of inhibition of α-amylase and α-glucosidase activity of nine marigold cultivars.

Bioactive Compound Contents	DPPH	ABTS	FRAP	α-Amylase Inhibitory Potential	α-Glucosidase Inhibitory Potential
Total phenolic content	0.652 **	0.498 **	0.759 **	0.612 **	0.766 **
Total flavonoid content	0.800 **	0.733 **	0.872 **	0.523 **	0.796 **
Total carotenoid content	−0.685 **	−0.365	−0.500 **	−0.866 **	−0.554 **
Lutein content	0.757 **	0.744 **	0.810 **	0.547 **	0.747 **

** Correlation is significant at the 0.01 level (2-tailed).

**Table 4 molecules-28-03314-t004:** Description of botanical entries.

Trade Name	Common Name	Scientific Name	Flower Color	Flowering Time
Rocco Deep Gold (RDG)	Marigold	*Tagetes erecta* L.	Orange	Apr–Jun
Golden King 001 (GK1)	Marigold	*Tagetes erecta* L.	Deep yellow	Apr–Jun
Yellow Queen 002 (YQ2)	Marigold	*Tagetes erecta* L.	Yellow	Apr–Jun
Nata 001 (NT1)	Marigold	*Tagetes erecta* L.	Yellow	Apr–Jun
Amari Deep Gold (ADG)	Marigold	*Tagetes erecta* L.	Deep yellow	Apr–Jun
Sara Deep Gold (SDG)	Marigold	*Tagetes erecta* L.	Deep yellow	Apr–Jun
Sara Orange (SO)	Marigold	*Tagetes erecta* L.	Deep orange	Apr–Jun
Amari (AM)	Marigold	*Tagetes erecta* L.	Orange	Apr–Jun
Angka (AK)	Marigold	*Tagetes erecta* L.	Deep yellow	Apr–Jun

## Data Availability

Not applicable.

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
