# Peer review of "In Vitro α-Amylase and α-Glucosidase Inhibitory Effects, Antioxidant Activities, and Lutein Content of Nine Different Cultivars of Marigold Flowers (Tagetes spp.)"

_molecules, 2023, doi:10.3390/molecules28083314_

Round 1

Reviewer 1 Report

The manuscript entitled "In vitro α-amylase and α-glucosidase inhibitory effects, antioxidant activities, and lutein content of nine different cultivars of marigold flowers (Tagetes spp.)" is a good work, however, there are plenty of areas needs to be improved prior to further consideration. Please find the suggestions below,

1. The abstract needs to be rewritten as it doesn't contain the background information of the study, major methods and quantitative results.

2. The introduction is Aldo seems to be oversimplified. Authors need to emphasize the diabetes and also associated secondary complications. Also  authors need to emphasize the possible drug targets and why they choose them (in detail, not in one or two lines).

3. What are the possible Mechanisms of antidiabetic activity of phytochemicals?

4 . What is the novelty of the present study. Authors should support their arguments with detailed literature on the marigold's anti-diabetic properties.

5. Authors can carry out more cell culture based studies (L6 myotubules) for better understanding of antidiabetic activity

6. The Discussion also should be detailed indicating the already proven anti-diabetic activity of Marigold

Author Response

Dear Reviewer,

Your response to the article titled “In vitro α-amylase and α-glucosidase inhibitory effects, anti-oxidant activities, and lutein content of nine different cultivars of marigold flowers (Tagetes spp.)” is greatly appreciated. On behalf of all the contributors, I've made the changes that a reviewer suggested.
Please open attachment

  1. The abstract needs to be rewritten as it doesn't contain the background information of the study, major methods and quantitative results.

I have corrected it.

  1. The introduction is Aldo seems to be oversimplified. Authors need to emphasize the diabetes and also associated secondary complications. Also authors need to emphasize the possible drug targets and why they choose them (in detail, not in one or two lines).

We add detailed literature on the diabetes and also on associated secondary complications (on lines 48 –55) and detailed literature on the marigold's anti-diabetic properties (on lines 84–86).

  1. What are the possible Mechanisms of antidiabetic activity of phytochemicals?

According to the findings of Kusmiati et al. [19] and Rodda et al. [20]. Both crude and lutein extracts from Tagetes Erecta are shown a significant effect on the blood glucose levels in diabetic rodent models. This implies that the lutein extract from marigold petals may block carbohydrate-degrading enzymes like α-amylase and α-glucosidase resulting in lower blood sugar levels.

4 . What is the novelty of the present study. Authors should support their arguments with detailed literature on the marigold's anti-diabetic properties.        

Marigolds have a wide genetic diversity which each cultivars will have different variations in the amount of bioactive compounds. In this study, the marigolds brought to the study were from Chiang Mai, Thailand, which had not been reported before. And most of the studies on marigolds have looked at their effects on visual function. There are few studies on the effects of marigold on diabetes. However, we add detailed literature on the marigold's anti-diabetic properties (on lines 84–86).

  1. Authors can carry out more cell culture based studies (L6 myotubules) for better understanding of antidiabetic activity

This study evaluated nine cultivars of marigold that had the highest content of bioactive compounds and an effective enzyme inhibitory effect. For future in cell culture and animal studies, the best-performing cultivars will be determined.

  1. The Discussion also should be detailed indicating the already proven anti-diabetic activity of Marigold

I have corrected it on line 260 to 269.

Reviewer 2 Report

General comments:

1. There are no "we", "our", etc. in scientific language. Please avoid use of personal pronouns as much as it is possible.

2. Unify terminology related to phenolic compounds. It should be always "phenolics" and not sometime "polyphenols" or anything else. Check a whole text.

3. DPPH is a radical while ABTS is a radical cation so not exactly the same. Both must be labeled in same and correct way through a whole text. Check and unify all.

4. I do not understand how/why authors express total phenolic content on quercetin as standard since it is flavonoids and it i used for TFC assay? In the literature TPC is always express on gallic, ferulic or chlorogenic acid. Please explain/check. this issue.

Specific comment:

Line 46: Flavonoids are phenolic compounds so can not be both here. Or you can add "in particular" in front of "flavonoids" here.

Line 65: Put "in vitro" term in Italic here.

Line 68: Specify here from where all varieties were obtained.

Line 80: What is now "GA" here? Check/correct/clarify.

Line 130: I think it should be here "is shown" not "showed"? In passive tense. Check/correct.

Lines 152 and 154: typos in words "total" and "negative". Correct.

Lines 158-161: I think that in this case negative correlation between total carotenoids and diabetic assays is actually good because you express your results based on IC50 values? If it is so it should be mentioned somewhere in the text.

Line 163: Correct "medicinal medicines" to avoid repeating of the almost same terms here.

Line 166: Rewrite/explain this "global biodiversity flower". It is incomprehensible here.

Line 230: How authors concluded this for "phenolic acids" here? Where did you determine content of these phenolic compounds? I did not find anywhere in the text. There is very nice and simple Arnoow's assay for determination of derivatives of dihydroxycinnamic acids. Please explain/add/delete.

Author Response

Dear Reviewer,

Your response to the article titled “In vitro α-amylase and α-glucosidase inhibitory effects, anti-oxidant activities, and lutein content of nine different cultivars of marigold flowers (Tagetes spp.)” is greatly appreciated. On behalf of all the contributors, I've made the changes that a reviewer suggested. Below are the details:

General comments:

  1. There are no "we", "our", etc. in scientific language. Please avoid use of personal pronouns as much as it is possible.

I have corrected it on line 223, 226, 241 and 378.

  1. Unify terminology related to phenolic compounds. It should be always "phenolics" and not sometime "polyphenols" or anything else. Check a whole text.

I have corrected it on line 193.

  1. DPPH is a radical while ABTS is a radical cation so not exactly the same. Both must be labeled in same and correct way through a whole text. Check and unify all.

I have corrected it on line 143, 238, 337 and 345.

  1. I do not understand how/why authors express total phenolic content on quercetin as standard since it is flavonoids and it i used for TFC assay? In the literature TPC is always express on gallic, ferulic or chlorogenic acid. Please explain/check. this issue.

I corrected that by analyzing the latest results. The results are provided using standard gallic acid, and are expressed as milligrams of gallic acid equivalent (GAE) per gram of dry weight.

Specific comment:

Line 46: Flavonoids are phenolic compounds so can not be both here. Or you can add "in particular" in front of "flavonoids" here.     

I have corrected it.

Line 65: Put "in vitro" term in Italic here.       

I have corrected it.

Line 68: Specify here from where all varieties were obtained.      

I have corrected it.

Line 80: What is now "GA" here? Check/correct/clarify.       

I have corrected it.

Line 130: I think it should be here "is shown" not "showed"? In passive tense. Check/correct.

         I have corrected it on line 17, 18, 20, 111, 118, 138, 149, 157 and 379.

Lines 152 and 154: typos in words "total" and "negative". Correct. 

I have corrected it.

Lines 158-161: I think that in this case negative correlation between total carotenoids and diabetic assays is actually good because you express your results based on IC50 values? If it is so it should be mentioned somewhere in the text.  

In term of linear correlation coefficients (r) between phytochemical contents and diabetic assays, I express the results based on percentage of inhibition. I've modified the name of table 3.

Line 163: Correct "medicinal medicines" to avoid repeating of the almost same terms here.  

I have corrected it.

Line 166: Rewrite/explain this "global biodiversity flower". It is incomprehensible here.

         I have corrected it.

Line 230: How authors concluded this for "phenolic acids" here? Where did you determine content of these phenolic compounds? I did not find anywhere in the text. There is very nice and simple Arnoow's assay for determination of derivatives of dihydroxycinnamic acids. Please explain/add/delete.

I have deleted it.

Round 2

Reviewer 1 Report

No more comments